# Brain Organoids: Filling the Need for a Human Model of Neurological Disorder

**DOI:** 10.3390/biology10080740

**Published:** 2021-08-02

**Authors:** Philip Jalink, Massimiliano Caiazzo

**Affiliations:** 1Department of Pharmaceutics, Utrecht Institute for Pharmaceutical Sciences (UIPS), Faculty of Science, Utrecht University, Universiteitsweg 99, CG 3584 Utrecht, The Netherlands; p.j.jalink@students.uu.nl; 2Department of Molecular Medicine and Medical Biotechnology, University of Naples Federico II, Via S. Pansini 5, 80131 Naples, Italy

**Keywords:** stem cells, 3D culture, disease modeling, tissue morphogenesis, neurodevelopment

## Abstract

**Simple Summary:**

This review article describes an overview of the developed advanced neural in vitro systems and the discovery of brain organoids including the recently developed improvement of this innovative technology. We also mention the main disease modeling applications of brain organoids and their potential impact in biomedical applications.

**Abstract:**

Neurological disorders are among the leading causes of death worldwide, accounting for almost all onsets of dementia in the elderly, and are known to negatively affect motor ability, mental and cognitive performance, as well as overall wellbeing and happiness. Currently, most neurological disorders go untreated due to a lack of viable treatment options. The reason for this lack of options is s poor understanding of the disorders, primarily due to research models that do not translate well into the human in vivo system. Current models for researching neurological disorders, neurodevelopment, and drug interactions in the central nervous system include in vitro monolayer cell cultures, and in vivo animal models. These models have shortcomings when it comes to translating research about disorder pathology, development, and treatment to humans. Brain organoids are three-dimensional (3D) cultures of stem cell-derived neural cells that mimic the development of the in vivo human brain with high degrees of accuracy. Researchers have started developing these miniature brains to model neurodevelopment, and neuropathology. Brain organoids have been used to model a wide range of neurological disorders, including the complex and poorly understood neurodevelopmental and neurodegenerative disorders. In this review, we discuss the brain organoid technology, placing special focus on the different brain organoid models that have been developed, discussing their strengths, weaknesses, and uses in neurological disease modeling.

## 1. Introduction

Neurological disorders are a group of disorders affecting the central and peripheral nervous systems. These systems include the brain, the spinal cord, cranial and peripheral nerves, nerve roots, the autonomic nervous system and the neuromuscular junction. These disorders include neurodegenerative disorders such as Parkinson’s disease (PD), and Alzheimer’s disease (AD); neurodevelopmental, or neuropsychiatric disorders, such as autism spectrum disorder (ASD), schizophrenia, depression, and anxiety; cerebrovascular disorders such as ischemia (stroke); headache disorders such as migraine or tension-type-headache (TTH); neuroinfectious disorders, including the infections of bacteria (mycobacterial tuberculosis), viral infections (Zika, HIV), fungal infections (Cryptococcus), and parasitic infections (malaria); brain malignancies, including primary or secondary brain tumors, and gliomas; and finally traumatic disorders, such as head trauma, or spinal cord injury.

Millions of people globally are affected by such disorders, resulting in them suffering daily from the consequences as well as the stigma and discrimination surrounding these disorders. Neurological disorders represent over 3% of the global burden of disease, and many of these disorders are ranked in the top 50 causes of disability-adjusted life years (DALYs) [1]. According to the WHO, hundreds of millions of people worldwide are affected by neurological disorders, counting more than 6 million people who die because of stroke every year, over 50 million people who suffer from epilepsy, and almost 50 million people who have dementia, among which 60–70% is thought to be a direct consequence of AD. The worldwide prevalence of migraine is estimated as highly as 10%, making it as big as 1/4th of the neurological burden [1,2]. 

Despite the high incidence and prevalence of neurological disorders, there remains a lack of effective treatment options for most of these disorders. This is likely due to different reasons. The first problem is lacking understanding of the underlying causes of the neurological disorders. This is caused by research models that poorly model neurological disorders. Additionally, a lack of access to diseased and healthy human material pre-mortem makes it difficult to study disorders in a relevant setting. Most studies into neurological disorders use either of two types of models to try and mimic the development of the disorders or determine the underlying causes and the pathology. 

The first type of models are in vivo animal models. Commonly, rodents are used due to the ease of use and low costs. Rodents are separated from humans via many millions of years of evolution, and their nervous system significantly differs to ours. Neurological disorders do not properly translate between humans and rodents as the genetic and environmental cues that cause a certain disorder in humans do not translate into a proper model of disease in rodents. Non-human primates (NHPs) and humans on the other hand have a closer, albeit not perfectly matched, genetic profile, but are expensive and difficult to work with. Non-human in vivo models have difficulty mimicking the disruption of social interaction that is common with neurological disorders, due to the lack of similar styles of communication between species.

Two-dimensional (2D) in vitro models are the alternative option to animal models. Commonly employed as a monolayer cell (co-)culture, this model gives researchers the ability to study the impact of potentially pathogenic mutations when working with human cells as opposed to animal cells. Shortcomings of this kind of model include the lack of cellular diversity and interconnectivity (especially in the highly regulated and interconnective 3D environment of the brain) in a 2D monolayer culture, as well as lacking proper presentation of disease pathology, development, and symptoms.

Recent breakthroughs in the fields of molecular biology and cell-genomics have shown that there is a large overlap in the genetic loci involved in the pathology of different neurological disorders. Techniques such as single-cell RNA sequencing (SC-RNA-Seq) have given researchers powerful options to pinpoint genetic hotspots for pathological mutations in the cells of patients affected by neurological disorders. A recent estimate has shown that more than 1000 pathological genetic mutations are commonly shared between a range of neurodegenerative and neurodevelopmental disorders [3,4]. This overlap provides researchers the understanding that most neurological disorders do not have a single (or even a few) fate-determining mutations, but rather a large and complex set of interactions between malfunctioning genes, the aging process, and in vivo-, and environmental factors [3,4]. High variance in symptoms between patients of the same disorder makes it difficult to make a timely diagnosis [5], and pinpoint all relevant environmental factors influencing the onset of these disorders without access to human models [6,7] and having done extensive studying of the disorders.

It is for these reasons that the need for an accurate human model to research the underlying pathological mechanisms of neurological disorders is greater than ever. Ethical concerns dictate the inability to work with live humans, and post-mortem tissue samples are usually at an end-stage of disease progression and pathology, if not aged. New technologies are being developed to allow working with human tissue, including an intact and accurate genetic code, while providing more in vivo-like cellular diversity and connectivity. Techniques such as CRISPR/Cas9 and other gene editing techniques allow us to introduce genetic mutations in the human genome in order to model mutations. Combining these options with the advances in brain organoid technology, researchers have access to complicated, available on-demand, ontogenically correct human models of neurological disorder development and pathology. 

Organoids are the next big step forward in the development of accurate in vitro models of disease. These 3D models account for many of the shortcomings of traditional 2D monolayer models in that they have higher cellular diversity, can form intricate networks of 3D spatial connections between cells, and develop much more similarly to the in vivo human brain. Human brain organoids are being rapidly developed and are currently being used to mimic brain and disease development and pathology. However, the brain organoid technology is not yet at a level where it can completely mimic all the interactions and structures of the developing human brain in vivo and more research needs to be done to ensure that the technology can be a completely accurate model for human neurological disorder. That said, the technology has already been used in conjunction with CRISPR/Cas9 to mimic a range of neurological disorders with promising results (see Figure 1) [8,9]. 

This review will focus on two topics; we primarily write about the development of advanced in vitro neural systems, the options available to a researcher, and the strengths and weaknesses of the different techniques and models. Secondly, we will focus on the application of different types of brain organoids as models of neurological disorders, with a specific focus on neurodegenerative, neurodevelopmental, and neuropsychiatric disorders. The review will be finished with an in-depth discussion on the strengths and weaknesses of the brain organoid technology and its current use in modelling neurological disorders, as well as a prospect for important research in the near future. The review will try to answer the question of whether human brain organoids can fill the need for a human model of neurological disorders.

## 2. Towards the Development of Brain Organoids

### 2.1. Pluripotent Stem Cells

To understand the underlying principles of advanced in vitro neural systems, it is required to understand the cells that form the functional core of this technique: pluripotent stem cells (PSCs). PSCs are cells that can differentiate into cells of all three major germ layers: the ectoderm, mesoderm, and endoderm. PSCs also have the ability to divide almost indefinitely. PSCs are commonly used in research in either of two ways. The first are embryonic stem cells (ESCs). These embryonic cells are derived from the undifferentiated mass of cells in the embryoblast [10]. Another kind of PSCs are so-called induced PSCs (iPSCs). iPSCs are cells that are induced in vitro using a range of techniques developed to turn terminally differentiated (somatic-) cells into pluripotent cells. Both ESCs and iPSCs are PSCs, i.e., cells that can differentiate into cells of all three germ layers. The cells produced in these germ layers are responsible for creating all the cells in the fetal and adult body.

iPSCs are primarily produced through a process called cellular reprogramming, a technique where somatic cells are exposed to a forced expression of the transcription factors Myc, Oct3/4, Sox2, and Klf4 via a retroviral vector [11]. Another way of producing iPSCs is through a technique called somatic cell nuclear transfer (SCNT). This technique includes the removal of the nucleus from a somatic cell and placing this nucleus in another cell (somatic or stem) from which the nucleus was removed [12]. This technique was popularized by mass media after the successful cloning of the sheep Dolly, however, it has since lost favor compared to the method of cellular reprogramming for a range of reasons including ethical complications and the difficulty of use compared to the transcription factor-based method. 

The benefits of using iPSCs extend beyond their ability to divide indefinitely or differentiate into different germ layers. iPSCs can be created using the genome of a patient with a certain disorder, and as a result, correctly mimic the genetic mutations of the disorder and model various pathways involved in the pathology and development of said disorder [13]. This ability is further enhanced by PSCs’ ability to self-organize differentiated tissues [14,15]. By using the patient’s own cells, iPSCs can be generated and used for transplantation therapy, as the genetic code and immunogenic profile will match the immune system of the host of the grafted cells [16,17,18]. 

### 2.2. Neural Stem Cells

To produce brain organoids, there is a need for PSCs to differentiate into cells of the ectoderm. The ectoderm includes most of the cells that we commonly find in the central nervous system (CNS), including cells of the brain such as neurons, astrocytes and oligodendrocytes, but also the peripheral nervous system (PNS), and epithelial cell lines [19]. Most cells from the CNS are derived from two types of multipotent stem cells, neural stem cells (NSCs), and epithelial stem cells [20,21]. NSCs are stem cells that are found in vivo in the developing neural tubes [22,23], and can produce so-called neural progenitor cells (NPCs). The most important of these NPCs are the radial glial cells, which can produce all different kinds of neurons (via basal/intermediate progenitor cells), as well as oligodendrocytes and astrocytes (via glial-restricted progenitor cells) [24,25]. 

In recent years, researchers have changed their opinions about the presence of renewable stem cells as a source of neurons and other cells in the adult CNS. Recent studies have shown that NSCs can produce NPCs in the adult CNS in various highly isolated regions, including the hippocampus, the olfactory bulbs, the neocortex, the septum, the striatum, and the spinal cord [26,27]. This has been shown primarily through experiments with rodent head trauma, and injury models in vivo, in which it was shown that growth-factor-dependent neurogenesis occurred primarily after CNS-related injury [28,29]. NSCs have also been shown to be useful for replacement therapy in rodent models of CNS-injury. Supplying NSCs to a site of injury for these rodents has been shown to improve neurogenesis [30,31]. NSCs are cells that are regulated by the microenvironment in which they are placed, and are capable of self-regulation [32,33]. NSCs and PSCs are the main starting cell-types for the formation of 3D neural structures in vitro.

### 2.3. Neurospheres

The first 3D culture system created in the pursuit of functional in vivo-like brain organoids were so-called neurospheres [34]. Neurospheres are clusters of cells cultured in the absence of an adherent substrate, floating in a compact group in suspension. Neurospheres are used to culture NSCs, and the NPCs that they create [35]. The main uses of this model are to study the NPCs in vitro, as well as assess the presence of the NSCs in the culture. Neurospheres are created using a technique called dual SMAD inhibition to ensure proper neural induction in free-floating conditions [36]. The culture medium that these neurospheres are grown in contain necessary growth factors for the cells to survive and proliferate, including epidermal growth factor (EGF), and fibroblast growth factor (FGF). The percentage of NSCs in a neurosphere is low, as these mostly contain NPCs [37]. Neurospheres are mainly used to perform a neurosphere assay (stemness assay), an assay designed to test for three major characteristics of NSCs. These characteristics include proliferation, self-renewal, and multipotency. Neurospheres can be dissociated by enzymatic digestion and cultured in single-cell wells. Self-renewal can then be assessed through the clonal analysis of the progeny. A small part of the single-cell wells will produce secondary neurospheres, indicating proliferation of the NSCs in those wells. Upon placing these secondary neurospheres in a medium containing differentiation-promoting growth factors, *multipotency* can be shown in the presence of terminally differentiated neurons, astrocytes, and oligodendrocytes [37].

The clinical applications of the neurospheres are simple but promising. The ability to produce NSCs from a patient’s own body provides options in replacement therapy. NSCs can cross the blood–brain barrier (BBB) [38,39] and integrate in the CNS of the host, without disrupting homeostasis or inducing a pro-inflammatory anti-graft reaction, to promote neurogenesis [40]. The high multipotency of NSCs can also be taken advantage of in a clinical setting by transplanting the NSCs into specific tissues, where they have shown successful differentiation and proliferation [41]. Studies into the therapeutic effects of NSC therapy have shown promising results for peripheral nerve regeneration [42] and auditory restoration [43]. The work of Dang and colleagues [44] has studied the effects of Zika virus infection in neurospheres on neural progenitor depletion. They found that the underlying mechanism was associated with the activation of TLR3, which is linked to the halting of neurogenesis and apoptosis. TLR3 inhibition proved to be effective in ameliorating the halted neurogenesis and inhibiting apoptosis.

Limitations of this method primarily include the sensitivity of the assay. The neurospheres’ microenvironment has an impact on the outcome of the produced NPCs. It has been shown that the progenitor cells produced in a neurosphere show differences when cultured under differing conditions. Cell density variations, differing constituents of media, growth and differentiation factor concentrations, and the protocol used for producing and maintaining the neurospheres have all been implied in differences in the composition of cell types and the properties of said cells in neurospheres. Additionally, the neurospheres do not mimic a perfect in vivo environment, and the known tendency of stem cells to react to the niche or environment [45] of their surroundings creates uncertainty about the ontogeny of the progenitor cells created.

### 2.4. Neural Aggregates

Another 3D culturing system that is similar to the neurospheres is the neural aggregate. Where neurospheres are made up of NSCs and the NPCs they create through differentiation and proliferation, neural aggregates are made up of PSCs [46,47,48]. The most common way to produce this kind of system is to first form an embryoid body (EB). EBs are three-dimensional aggregates of pluripotent cells, which can differentiate and result in the development of tissues of all three germ layers [49]. PSCs are capable of differentiation and morphogenesis, resulting in the development of so-called microtissues. In the case of neural aggregates, stem cells are pushed to (primarily) differentiate toward the ectoderm by Dual SMAD inhibition [50,51]. Microtissues manifest in the shape of developing buds from the EB, which go on to form a neural tube-like structure [46,47,48]. Within these developing neural tubes, NSCs and NPCs are present, which undergo differentiation and proliferation similar to an in vivo developing embryo.

Similar to the neurospheres, this model can be used to assess the presence and qualities of NSCs and the NPCs in their progeny. More unique to this model is the possibility to create specific NPCs and remove them from generated neural tubes for analysis and culturing. Additionally, the neural aggregates provide a better spatial–temporal overview of the development of the neural tube and the NSC progeny thereof [46,47,48]. The neural aggregates also provide a nice model for embryogenesis and neurogenesis. Due to the self-regulation of forming neural tubes in the neural aggregates, the neural aggregate method forms a more robust protocol compared to the neurospheres [52]. A study by Colombo and colleagues [46] has used neural aggregates to study neurite growth and attachment in the rodent brain after transplantation. They found that the NSCs maintain those abilities after being grafted. Another study by the group of Ahn and colleagues [52] concluded that neural aggregates are suitable for the study of tauopathies and neuropathogenic processes seen in prion disease. Additionally, they determined drug efficacy results similar to the results found in in vivo studies.

A major limitation for both neurospheres and neural aggregates comes from the limited size the models can reach. Both neurospheres and neural aggregates will go apoptotic near the center of the model due to the lack of gas exchange and waste metabolite removal [53]. Additionally, both the neurospheres and the neural aggregates can only be used to answer simple research questions, as they lack the complicated in vivo environment that stem cell behavior is regulated by. As these models consist almost completely of NSCs and NPCs, they lack in cellular diversity (neurons, glia), and intercellular connectivity. Newer 3D models should require not only the inclusion of relevant cell types for the CNS, but also represent the brains in vivo 3D structures with cortical layering, and clearly separated brain areas, with area-specific cellular heterogeneity and connectivity.

### 2.5. Neural Rosettes

Whilst a 2D culturing system, and therefore not factually an organoid, neural rosettes are morphologically identifiable monolayer cultures that are designed to mimic the developing neural tube, as seen from a sagittal viewpoint [54,55]. Simply put, neural rosettes are the developmental signature of NPCs in cultures of differentiating PSCs. Neural rosettes contain NSCs, which surround a central lumen in the form of a tube (similar to the neural tube). The NSCs surround this central lumen and are undergoing the mitosis at the luminal side. For a 2D culture system, the neural rosettes accurately mimic the neurogenesis [54,55]. The creation of neural rosettes typically relies on the formation of embryoid bodies from PSCs. Following the creation of the embryoid body, the medium will be changed to a neural induction medium. After culturing the developing NSCs, a neural rosette can be selected in the culture and replated for the further culturing of the NPCs [55,56]. The neural rosette model is used for neural induction, i.e., the production of specific terminally differentiated neurons and glia, or their precursors (NPCs). Researchers prefer this induction method over models such as neurospheres and neural aggregates because the 2D structure allows for the easier removal of waste and gas exchange. This means that the model does not go apoptotic as the simpler 3D models do, and allows for neural rosettes to stay in culture for a longer period of time [57]. This in turn ensures bigger cellular diversity, and more terminal cellular differentiation. After the neural rosette forms, the NPCs in the 2D culture can be expanded and differentiated into any of its more mature progeny cell lines, with phenotypes resembling those from many regions across the CNS [58,59]. Further benefits of this model include the ease of use and maintenance, low labor intensity and low costs.

Limitations of the rosette model are much in line with the traditional shortcomings of 2D monolayer cultures. The two biggest problems include upscaling difficulties, and a lack of cellular interconnectivity. Upscaling difficulties arise from the monolayer aspect of the model. 2D models do not provide researchers with a lot of material to work with, and as the neural rosette is adherence dependent, upscaling requires a lot of space, labor, and costs. The lack of cellular interconnectivity also arises from the fundamental characteristics of a monolayer culture. Whilst cellular diversity is better in neural rosettes, the lack of interconnectivity between cells in this model dictate that its primary use is in neural induction protocols, as opposed to studying neural circuit functioning and formation.

## 3. Complex 3D Organoids and the Use of Patterning Factors

Neurospheres and neural aggregates can be used to study research topics such as NSC behavior, proliferation, differentiation, and self-renewal, as well as the simplified formation of neural tubes, and the presence and authenticity of NPCs. Because these models are all made up (almost) completely of NSCs and NPCs, and the lack in cellular diversity of neurons, glia, and oligodendrocytes, a need for more diverse models exists. For more advanced models, it is not only important to mimic cellular diversity, but also cellular interconnectivity, and 3D structure of the CNS (e.g., cortical layering, regional interconnectivity). More modern brain organoids, which will be discussed below, have been shown to produce structures that are organized in a manner similar to the CNS in human embryos and fetuses [60,61]. The cellular diversity, organization, and connectivity can be regulated by addition of so-called patterning factors. A list of important properties to develop and regulate in more advanced brain organoids include:Organoid functionality: the ability of the organoid to mimic the functionality of the modeled brain area. For pituitary-specific organoids, this means inducing changes in hormonal secretion and balancing. For organoids that include dopaminergic neurons, such as midbrain-specific organoids, this includes the ability to synthesize dopamine from L-DOPA and the ability of its neurons to use it as a neurotransmitter.Cellular diversity: the induction of all relevant cell types within the organoid of the modeled brain area. Most areas of the brain have a large variety of cells, but the final functionality is decided by the cells within the highly specific areas. For every modeled system, researchers need to ensure and quantify the presence of the right types of neurons, astrocytes, and oligodendrocytes. The presence of microglia, and epithelial vascularization are also important and will be described in more detail in later sections. Distribution and ratios between are vital for the normal physiological functioning of the organoids.Connectivity between cells: accurate and functional connections between cells (especially neurons) is highly important for creating a good model of the CNS in vitro. Organoids that have only excitatory or inhibitory neurons will poorly model the human brain since many of its areas (most notably the cortex) have cells in different layers that relay signals and communicate with each other.Development of the modeled area: proper developmental profiles of the organoid are important to guarantee the maturation of the cells and tissues. This is necessary for ensuring the proper functionality of, and the interactions between the cells in the organoid.Structural integrity: many areas in the CNS have highly specific structural integrity and shape which adds to the functionality and connectivity of the CNS. Most notably, areas such as the hypothalamus, cortex, and retinal cups have structures that are fundamental for the proper functioning of the CNS.

Patterning factors are growth factors and medium supplements that can be used to improve the cellular diversity, organization, differentiation, and connectivity of brain organoids. Typical patterning factors for brain organoids include molecules and proteins such as bone morphogenic proteins (BMPs), Wnt, Sonic Hedgehog protein (Shh), retinoic acid (RA), and FGF [62,63]. The complexity of the organoid can be regulated by adding or omitting these exogenous patterning factors. Adding more patterning factors can allow for the production of “single-region brain organoids” (e.g., cortical spheroids) [64]. Limiting the exogenous patterning factors can lead to the so-called “whole-brain organoids”, or cerebral organoids. Whole-brain organoids include representative structures and markers typical of different brain regions whereas region-specific organoids that only show the features of a specific region of the CNS [64]. 

Additionally, patterning factors can be used to increase organoid reproducibility, which is a big problem across more modern organoid models (see Figure 2). This allows for more reproducible and comparable results from batch to batch as well as within and between laboratories and researchers [65,66]. 

## 4. Single-Region Brain Organoids

Single-region brain organoids differ from whole-brain organoids since they are used to model specific regions of the CNS. These areas can be as large as parts of the hind-, mid-, and forebrain, but are generally smaller and more specific, such as models of the pituitary glands. Through the addition of exogenous patterning factors and the strict control of the induction environment (the niche), NSCs in the neural aggregates can be made to differentiate into highly specific and accurate models of certain areas of the brain.

The precursor to most single-region brain organoid models was the so-called serum-free floating culture of embryoid body-like aggregates (SFEB), developed by Watanabe and colleagues [67]. The more modern version of this model, named SFEBq (the q stands for quick reaggregation) was developed by Eiraku and colleagues [68], and is commonly used as a starting point for the induction of many single-region brain organoid models. In SFEBq, the PSCs in the neural aggregates differentiate and form a neuroectoderm-like epithelium. This epithelium can be used to generate cortical neurons or other NPCs to start the development of complex 3D organoids in vitro [69]. Prof. Pasca’s group built on this method and developed neural structures called cortical spheroids [70]. These cortical spheroids were developed without the extracellular matrix (ECM) and using limited patterning factors, and relied for a large part on self-regulation to develop cortical structures. The cortical spheroids uniquely presented with both superficial and deeper neurons, and the presence of a rare type of astrocyte to generate the quiescent astrocytes in vitro was confirmed [71]. The lack of patterning factors used to create the cortical spheroids illustrate that these factors are mainly needed to specifically tailor non-cortical organoids.

Building on the SFEBq model, many different kinds of specific brain organoids have been designed in the literature. Examples of single-region brain organoids that we will discuss include organoids of the hypothalamus [72], cerebellum [73], and midbrain [74].

A hypothalamus organoid was produced by Wataya and colleagues [72]. They induced neuroectodermal progenitors from ESCs. The complete removal of exogenous patterning factors during early differentiation led to the efficient generation of rostral hypothalamic-progenitor cells (Rax+/Six3+/Vax+). Insulin was removed from defined growth media to generate these progenitor cells, due to involvement of inhibiting the Akt-dependent pathway through which these cells are generated. These progenitor cells generate Otp+/Brn+ neuronal precursors which subsequently turn into magnocellular vasopressinergic neurons that are capable of efficient hormone release. The differentiation is induced via the incubation of the progenitor cells with Shh protein. The authors concluded that ESC-derived neuroectodermal cells can spontaneously differentiate into hypothalamic progenitors, which generate hypothalamic neuroendocrine neurons. Muguruma and colleagues [73] used a complex regimen of patterning factors to induce Purkinje cells in an organoid from human ESCs. Purkinje cells are GABAergic neurons in the cerebellum. The study found that a polarized cerebellar-like structure self-organizes in a human ESC culture. The neuroepithelium from which the cerebellar structure originates differentiates into electro-physiologically functional Purkinje cells through the addition of the fibroblast growth factor-19 (FGF-19). FGF-19 promotes the induction of a polarized neural-tube-like structure that can form the cerebellum. Furthermore, the addition of SDF1 and FGF-19 promoted the generation of a continuous cerebellar plate neuroepithelium, similar in structure to the embryonic developing cerebellum. A study by Kim and colleagues [74] was performed to model the impact of the G2019S mutation in the LRRK2 gene on the phenotype of PD in a 3D midbrain organoid. The study showed that the mutation can produce 3D pathological hallmarks observed in patients with this specific mutation. Additionally, the study described performing the analysis of the protein–protein interaction network in the mutated organoids, and found a role for thiol-oxidoreductase (TXNIP), and showed it to be functionally important in the development of the PD phenotype caused by the G2019S mutation. Midbrain modeling was also performed in a work from Prof. Ng’s group, in which they generated midbrain-specific organoids [75]. They created midbrain-specific organoids called midbrain-like organoids (MLOs), in which they detected electrically active, and functionally mature, dopaminergic neurons.

## 5. Whole-Brain Organoids

The term whole brain organoid is used for organoids that model several areas of the CNS. This approach aims to model interconnectivity between the brain regions, development for larger areas of the cerebrum, and structural integrity at a larger scale. 

The first cerebral organoid model was created by Lancaster and colleagues [76] in 2013. They generated the first single neural organoid that modelled multiple areas of the brain. Unlike single-brain-region organoids, the whole-brain organoid method uses limited exogenous patterning factors, instead relying on endogenous factors and the self-organizing ability of the NSCs. In this technique, neural aggregates, which form the basis of the more advanced 3D model of organoids, are embedded in an ECM Corning Matrigel. The ECM induces the polarization of the neural progenitor sheets, and supports the outgrowth of neuroepithelial buds. These buds can then differentiate and spontaneously develop into the different brain regions without the addition of exogenous patterning factors [76,77]. The study also noted that the brain regions contained layers of developing neurons that model the development and organization of the developing human brain. The whole-brain organoids mimic the early CNS developmental program closely, and additionally can be used to model human brain diseases. Another benefit is that the organoids can stay in culture for more than a year, allowing studies on the survival and maturation of neurons and the developing brain.

A study by Raja and colleagues [78] modeled the (AD) phenotype in an organoid induced from a patients’ iPSCs. The result was a self-organizing organoid that modeled Aβ aggregation, tauopathies, and other symptoms of AD in an age-dependent manner. They concluded that the model could be used to test pre-clinical drug efficacy.

## 6. Limitations of Complex Organoids

Although the modern brain organoids are a big step forward when it comes to their applications in research and the modeling of neurological disorders (See Table 1), they still have some limitations. The first limitation is in the fact that the organoids do not develop past the stage of a pre-natal brain [76]. This is likely due to the fact that the organoids do not develop a vascular system to aid with nutrient circulation, gas exchange and waste removal (see section on vascularization). This additionally leads to apoptosis in the core of the organoid, which results in a challenged environmental niche for the stem cells, and in line with this dysregulated development and differentiation [79]. Another problem would be the lack of cellular diversity and interconnectivity of cells within-, and between different brain regions. The fundamental self-regulation of these organoids has been shown to favor specific cell profiles over more diverse ones, leading to a lack of certain cell types. An example of this is the imbalanced ratios of excitatory and inhibitory neurons (see *cellular interconnectivity*). In some organoids, the balance between these two types of cells does not accurately reflect the situation in vivo [80]. 

## 7. Microglia and Cellular Diversity

Cellular diversity in the brain is highly important for proper physiological functioning. Many different subtypes of cells exist in the brain, and each of these subtypes have their own unique properties. They fill and complement the needs of the environment in which they reside. The main cell types in the brain include neurons, astrocytes, oligodendrocytes, microglia, pericytes, and a small amount of NSCs and NPCs. These cells play an important role in the functionality of the CNS, but also have complex parts in maintaining CNS homeostasis, and can affect disorder pathology. The health of the CNS relies on its cells’ ability to complement each other and promote overall survival and proliferation. To date all of the listed different subsets of cells can be developed within brain organoids (see Figure 3).

In the brain organoids models we described, all cells were derived from PSCs that were made to differentiate into the (neuro-)ectodermal germ layer. However, the CNS does include cells from different germ layers that are fundamental for normal CNS development, physiology, and homeostasis, and can affect disorder pathology. The most common of such cells are microglia.

Microglia are commonly known as immune cells in the CNS. These cells are not derived from the ectoderm as other cells of the CNS are, but from the mesoderm [113,114]. Among cells that originate from the mesoderm are blood cells, which originate from a process called hematopoiesis. Microglia are created during an early stage of embryonic development, during a process called primitive hematopoiesis. This process occurs in the yolk sack, where erythro–myeloid–progenitor cells (EMPs) are created, which migrate to the developing neural tube [113,114]. There, the early microglia will keep developing together with the developing CNS, gaining new properties and functions as the CNS develops and matures. The microglia in the brain have been shown to be self-renewing and capable of unending proliferation [113,114]. Microglia are involved in more than just the immune system. They have been shown to be responsible for aspects of CNS homeostasis. Neuronal survival in the developing brain is supported by the release of growth factors like insulin-like growth factor-1 (IGF-1) and fractalkine (CX3CL1) among others [113]. It is accepted that microglia have a key role in supporting neuronal development and survival. Additionally, microglia have been shown to be involved in the process of neuronal apoptosis [113]. Other roles of microglia include synaptogenesis, antigen presentation, and the induction/inhibition of proinflammatory pathways through cytokine and chemokine release. 

Microglia have also been shown to be involved in many neurological disorders [113,114]. Microglia have activation states that are triggered by pathogenic infections or CNS injury that can make their behavior induce a proinflammatory response. The different activation states have been shown to have different behavioral profiles, and are an important inclusion in a neurological disorder model [113,115]. The generation of in vivo representing microglia has proven a tough challenge, with the current predominant method used being the induction of a yolk-sack like structure to mimic primitive hematopoiesis [115,116]. The generation of a whole-brain organoid with a self-renewing microglia population is challenging, as the microglia and the organoids both require a complicated and controlled environment to successfully develop from different germ layers.

A recent study from Ormel and colleagues [117] has shown innate microglia development in a cerebral organoid based on a protocol from Lancaster and colleagues [76,77]. The transcriptome, morphology, and functionality of the organoid microglia were similar to those seen in adult microglia from a post-mortem brain. They made the hypothesis that the induction of the organoid neuroectoderm, which does not rely on the traditional dual-SMAD differentiation protocol but rather embedding in a Matrigel, does not remove all progenitors from the mesoderm. These mesodermal progenitors could develop into functional microglia in the developmental niche of the cerebral organoid. Another study by Park and colleagues [118] created an organoid model for the study of AD, where they included microglia and worked with a 3D microfluidics system. The organoid accurately modelled pathologies from AD, and microglia recruitment, and neuroinflammatory-, and neurotoxic behavior were similar to in vivo. They concluded it could be used as a starting point to develop more complex organoids.

## 8. Cellular Interconnectivity

One of the most unique aspects of the brain is in its ability to process and distribute information. Neuronal circuits are at the basis of this functionality in the CNS and are widely spread throughout the brain. All areas of the brain have circuits responsible for specific functions, be it memory, speech, breathing, posture, or anything else. Many of these abilities are linked to each other through the formation of neuronal circuits in the brain. Making a memory of something you see is very common, but the centers for memory (hypothalamus, amygdala, neocortex), and vision (thalamus, visual cortex) are not the same. This process relies on the formation of neuronal circuits within-, and between areas of the brain. This ability of the brain to process information obtained by one part of the brain in another area is arguably one of its strongest features, and is especially reliant on the connections between cells, specifically neurons, and entire areas of the CNS.

Interconnectivity between neurons is mimicked more accurately in complex brain organoids, as opposed to 2D culture alternatives, which do not have the spatial area possible to form intricate 3D networks of neuronal connections. Making sure that areas of the organoids are connected to each other via the formation of neuronal circuits is highly fundamental in creating a brain organoid that models the in vivo CNS.

A common occurrence in complex organoids is the mismatching of neurons and formation of non-functional neuronal circuits. The most common example of this is a disruption in the ratios of inhibitory and excitatory neurons, which has been shown to be imbalanced in complex organoids [80]. This imbalance is usually linked to patterning factors that induce dorsal-, or ventral organoids, and as such, is primarily an issue in single-region brain organoids. ‘Dorsalizing’ patterning factors used to induce an organoid model of the cerebral cortex (dorsal telencephalon) will favor excitatory neurons, where ‘ventralizing’ patterning factors will induce a model of the ventral telencephalon with an abundance of inhibitory neurons [80]. A possible solution to solve this issue of neuronal/cellular imbalance is the organoid fusion technique.

Fusion organoids are the result of the co-culturing of organoids with two different identities. An organoid modeling a dorsal telencephalon and an organoid modeling the ventral telencephalon could be co-cultured to permit a possible fusion of the organoids. A technique such as this would enable the creation of organoids with migrating inhibitory and excitatory neurons, and allow for the construction of local inhibitory–excitatory neuronal circuits. Pasca group [110] pioneering work achieved to fuse organoids of the pallium and the sub-pallium together in an attempt to form a so-called assembloid, modeling the migration of neurons between these two areas and the formation of a neuronal circuit. Another application of the assembloid technique is to co-culture organoids modeling two different brain regions, allowing for models with the size of whole-brain organoids and the complexity of single-region brain organoids. Examples of this application have been achieved for cortico-striatal [119] and cortico-motor [120] assembloids.

## 9. Vascularized Organoids

One of the major problems most organoid models currently face is the inability to grow beyond a certain size. The most complex brain organoid models grow to the size of a fetal brain a couple of months in vivo [64,65]. The main reason development stops at this point is the lack of a vascular system, which normally develops from the mesoderm. Most neuroectoderm induction protocols remove all cells from the mesoderm, and so the vascular system is not formed. The lack of circulation results in a lack of oxygen and gas perfusion as the size of the organoid increases. Additionally, toxic waste metabolites from cells are not removed from the system, and nutrients from the growth medium have a harder time reaching the cells at the core of the organoid [121,122]. As a result of these issues, the organoids grow apoptotic at the center, and cell death will occur. This will lead to stunted growth of the organoid, and impaired maturation of the cells [121,122]. The introduction of mesenchymal stem cells or iPSC-derived endothelial cells could help in the formation of a vasculature system [121].

Similarly to the inclusion of microglia, a system has to be developed to allow for the development of the mesodermal vasculature. A study by Pham and colleagues [123] has shown that the inclusion of human endothelial cells derived from a patient’s own iPSCs in an organoid produced from those same cells can induce the formation of a vasculature system. Additionally, in vivo, they showed that the transplantation of a human organoid with vasculature into the brain of the mouse showed the survival of the organoid post 2 weeks, whereas the organoid without vasculature did not survive over the same time period.

A study by Cakir and colleagues [124] has modified a protocol for human cortical organoids (hCOs) to form a functional vascular-like system. They engineered human ESCs to ectopically express human ETS variant 2 (ETV2). Cells that express this gene contributed to the formation of a complex vascular-like networks in hCOs. They found promising results suggesting that the vascular-like system helped in functionally maturing the organoid. Additionally, they found that the vascularized organoids received properties of the blood–brain barrier, including nutrient transport, tight junction protein expression, and more. In vivo results showed that ETV2-expressing endothelial cells supported formation of a vascular system, resembling the early prenatal brain.

## 10. Advances in Bioengineering

The field of organoids is rapidly moving towards a state where common practice dictates that 3D cultures are combined with bioengineering techniques. Bioengineering can help solve some of the biggest remaining problems in the organoid field. These problems include:Reproducibility of organoids;Gas exchange and nutrient delivery;Negative impacts of the culturing environment (stress);The need for an adherent growth platform;An extracellular matrix to enhance growth and cellular functioning;An excess of mesodermal and endodermal tissue.

Matrigel is a substrate made of gelatinous protein mixture that is used to culture cells [125]. Specifically, Matrigel is used to mimic the complex extracellular environment in which cells grow in vivo. It has been shown that cells do not express the same behavior when cultured on plastic, as they do in vivo [125]. Culturing on Matrigel does allow for these complicated behaviors, an example of this behavior would be the formation of vascular tubes by endothelial cells growing on a Matrigel, required to form a circulatory system. Many brain organoid protocols have used Matrigel to provide the organoid with both a basement cell culture platform, as well as to model the extracellular environment since the whole-brain organoid protocol from Lancaster and colleagues [76]. 

Polymer-based scaffolds can be used to support organoid growth. They can be given shapes to allow for the precise control of the formation and growth of organoids. Scaffolds can be used to increase the amount of neuroectodermal tissue available for the organoid to grow from. Current protocols that do not use patterning factors have issues controlling the amount of mesodermal and endodermal tissue generated [64,126]. Prof. Knoblich’s group clearly showed the value of implementing scaffolds in the generation of brain organoids. Indeed, a recent work from this group proved that poly(lactide-co-glycolide) copolymer (PLGA) fibers can be used as a floating scaffold in order to elongate embryoid bodies. The so-called microfilament-engineered cerebral organoids (enCORs) show a more consistent cortical development, thus proving that biofabrication approaches can significantly improve the quality and reproducibility of brain organoids [60].

Another use of scaffolds is the ability to gain spatiotemporal control over the growth of the organoid [65,126]. This leads to more reproducible organoids that show less batch-to-batch differences and allow for better analysis. Scaffolds additionally in the future might allow for the precise control and release of patterning factors, as well as guiding PSC behavior to induce a single neural tube-like structure [65].

Spinning bioreactors or miniature bioreactors are reactors made to size for culture of (a single) organoid(s). Their main use is to increase gas exchange and nutrient delivery for the organoid in culture by spinning the organoid around [65,127,128]. Whilst the size of the organoid is still limited by the absence of a circulatory system and the size of the bioreactor, the core of the organoid will not go necrotic as quickly [65,127,128]. Additionally, bioreactors are also useful for reducing the stress and impact of the culturing environment of the organoids. Shearing especially can lead to negative effects on the growth and development of the organoid in culture.

Qian and colleagues [129] developed protocols for the development of fore-, and mid-brain specific organoids, as well as hypothalamic organoids. Using miniaturized spinning bioreactors to ensure proper nutrient delivery and gas exchange, they created fore-brain specific organoids to model Zika virus exposure in a complex system. They found that both the African and Asian zika virus strains preferentially infected neural progenitors in the developing human brain. The infection leads to cell death, and reduced proliferation, leading to a neuronal cell-layer with reduced volume, similar to microcephaly.

## 11. Discussion

The brain organoid technology has come become popular over the last two decades as a powerful and unique tool to study the development and working of the complex CNS. Brain organoids have been extensively used to study the developing CNS, as well as the disorders that are associated with it. To answer the research question “Can brain organoids fill the need for a human model of neurological disorders?”, we need to weigh the strengths and limitations of brain organoids and discuss whether future enhancements will succeed in making brain organoids a viable human model of neurological disorders.

To define a good model of neurological disorders, we first define the properties of such a model. A model of neurological disorders should be usable to study the onset, the development, the pathology, and the causes of the disorder in an accurate way. Furthermore, a good model allows researchers to look at the inner workings of the disorder, including malfunctioning pathways, abnormal protein interactions, and pathological mutations. Finally, the model should allow for testing with drugs or molecules to try and ameliorate tissue homeostasis and normal functioning, if not cure the disorder. To assess whether a new model is successful, the shortcomings of previous models should be defined. Previous models for neurological disorders included animal models, 2D monolayer cultures, post mortem human tissue biopsies, and live humans. Each of these models had one or more major shortcomings, and if organoids can be used to improve on each of these issues, we can consider them an improvement, and potential replacement for the previous models. 

Many in vivo animal models used to study neurological disorders have failed to mimic the onset of those disorders. The onset of neurological disorders is related to environmental cues and genetics. These disorders have been shown to underly a complicated web of interacting pathogenic mutations. Due to this, the onset of these disorders cannot be accurately mimicked in animals, as the genes involved in onset of similar disorders are not the same. Furthermore, the development and pathology of the disorders also does not accurately model that in humans. Finally, drug testing in animal models has often not proven successful as the toxicity, clearance, and efficacy of drugs from and in the CNS do not translate well from animal model to humans due to the blood–brain barrier differences. 

2D monolayer cultures are unsuccessful when trying to model the complicated 3D in vivo environment of the CNS and lack the interconnectivity that comes with such a 3D structure. Whilst the models are good for testing drug–protein interactions, they fail to properly show the efficacy and clearance of the drugs. Studying the onset, and development of the disorder is also difficult in 2D cell cultures. 

Post-mortem human tissue biopsies give researchers human material from diseased patients, ensuring a proper genetic code. These tissues can be used to create cell cultures or to perform other analytical studies. Shortcomings of this model include a lack of access to on-demand high quantities of material, the highly aged state of most samples (which is especially bad when trying to model neurodevelopmental disorders, but better for neurodegenerative disorders), and the fact that most of the material is at an end stage of disease, making it impossible to study disease onset and development in healthy tissue. Moreover, studying neurological disorders in live humans is very complicated, as the CNS is poorly accessible without performing invasive surgical procedures. 

Brain organoids have been used to successfully study neurological disorders and make up most of the shortcomings of the previously mentioned models. The ability to generate brain organoids from the cells of human patients with neurological disorders allows researchers to accurately model the genetic background of the patient in a complex 3D environment with endogenous signaling, similar to an in vivo model. Unlike animal models, brain organoids have been successfully shown to model neurological disorders when generated from cells carrying the mutations required for onset of the disorder. The brain organoid models have shown similar pathology to the in vivo human [81], and testing for the toxicity and efficacy of drugs (but not clearance and dosing) can be performed and translated to the human body [130,131]. Examples of the possibility to use brain organoids to model acute toxicity has been recently given both in Zika virus [132] and Parkinson’s disease [100] modeling. Brain organoids form far more complex structures, accurately mimicking the neural circuits found throughout the developing human brain [133,134]. Because organoids are derived from PSCs, the aged status and the diseased gene profile of the post-mortem human brain do not affect the organoid models of neurological disorders. Finally, unlike live humans, work with organoids does not carry similar ethical considerations, and culturing them is far easier than working with live humans.

It must also be mentioned that brain organoids do have limitations in their current state. The main limitation of brain organoids is clearly their divergence from the human brain development that eventually still lead to several differences between the two systems. Compared to the in vivo developing human brain, organoids lack in cellular diversity and connectivity between neurons and areas of the brain [135,136]. Additionally, the balance between many different kinds of brain cells is not similar to that of the in vivo human brain. Excitatory and inhibitory neurons are often dysregulated in organoids, and the different types of neurons and astrocytes are generally dysregulated [137]. Additionally, most brain organoids are induced through the overexpression of the ectoderm and inhibition of the mesoderm and the endoderm. This means that cells such as microglia, which have key roles in immune regulation, neuron maturation, and brain homeostasis, are not present. Subsequently, this also means that organoids lack vascularization, as part of the blood circulation is derived from the mesoderm. This results in a lack of gas exchange, waste removal, and nutrient transport across the organoids above a certain size. As a result of this lack of vascularization, brain organoids do not age (in size and developmental profile) past a developmental stage a couple of months after conception. The lack of circulation also suggests that drugs are not spread throughout and cleared from the brain in a normal fashion [138,139]. 

These issues are not unsolvable, however. The implementation of endothelial cells in the developing organoids has been shown to lead to the development of an elementary circulatory system, albeit with questions on the efficacy of the blood circulation [123,124]. The added circulatory system could open up possibilities in drug clearance-, and dosage-testing. IPSC-derived microglia have also been added to organoids in an attempt to create a self-renewing microglia population in the organoids, to study the effects of microglia in the developing fetal brain, as well as their impact on disorder pathology [140,141]. The induction of the mesoderm in organoids has been studied to attempt to induce the development of microglia and circulatory systems [142,143]. Assembloids have been created to study the migration of neurons, as well as ameliorate the imbalance of neuronal subtypes between organoids [144]. Future research could see us fuse multiple single-region brain organoids of different regions of the brain together to create larger and more complex whole-brain organoids, although the presence of a circulatory system is mandatory in achieving this.

## 12. Conclusions

Brain organoids can be used to study neurological disorders and the impact of genetic mutations and environmental factors on the onset, development, and pathology of these disorders. They can be used to study the inner working of cells affected by the disorders, including altered pathways, protein interactions, and mRNA expression. Brain organoids can be used to perform drug testing, allowing us to accurately model drug efficacy, and in the future, clearance and dosing studies in a complex 3D human system. Brain organoids have been used to study neuronal connectivity, the neurodevelopment of the brain and the maturation of its cells, neuro-immune interactions, viral transduction, and the transcriptomics and epigenetics of cells in healthy and diseased brains. Evidence from all of the studies mentioned in this review seems to suggest that brain organoids make excellent models of neurological disorders, and that they overcome most of the shortcomings of previous models of neurological disorders. 

To further improve on the concept of using brain organoids as models of neurological disorders, researchers will need to ensure that they accurately resemble the in vivo CNS, by ensuring the following points:The controlled induction of the mesoderm to allow for the development of a vascular system, and ensure the presence of microglia. This is necessary to allow organoids to grow past the developmental stage of a couple of months post conception.Balancing the ratios between neurons, astrocytes, and microglia to ensure proper cellular maturation, interconnectivity, and homeostasis.The improved spatiotemporal control of the development and differentiation of the organoids, using tools derived from bioengineering such as scaffolds and extracellular matrix to ensure the controlled delivery of patterning factors and the induction of the correct germ layers.

If researchers can make these three key aims a reality within the field of organoids, they can potentially be used to fill the need for a human model of neurological disorders.

## Figures and Tables

**Figure 1 biology-10-00740-f001:**
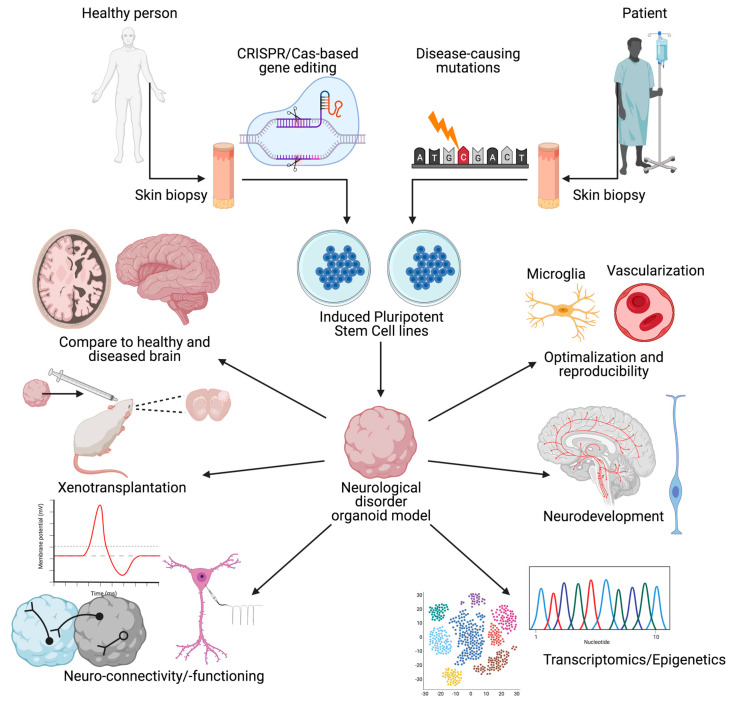
A schematic overview of the creation of and uses for organoid models of neurological disorders. Overview of the different research outcome measures and tools available to researchers working with brain organoids.

**Figure 2 biology-10-00740-f002:**
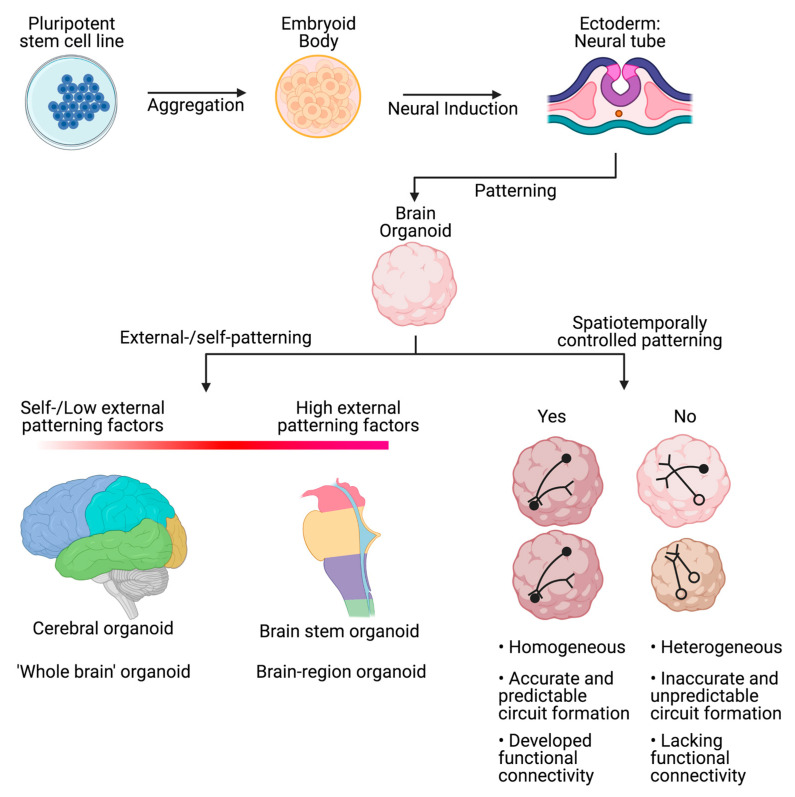
Schematic overview of the differentiation progress of organoids with patterning factors. Outcome differences between the use of high or low concentrations of patterning factors and the spatiotemporal control of patterning.

**Figure 3 biology-10-00740-f003:**
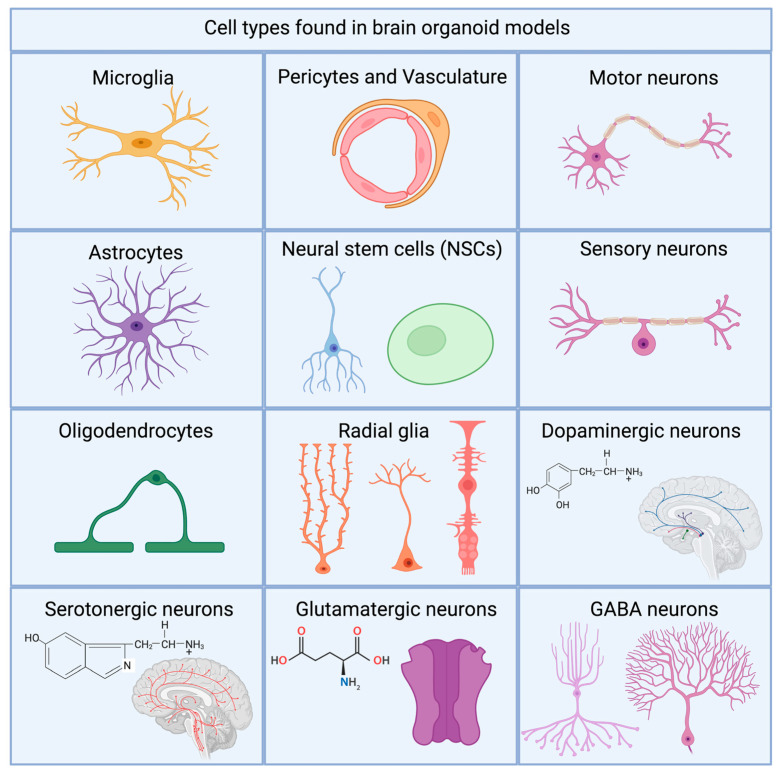
Non-exhaustive schematic overview of the different types of cells found in different organoid models.

**Table 1 biology-10-00740-t001:** Overview of the different diseases/disorders/syndromes that are modelled in complex organoids.

Disease/Disorder/Syndrome	Organoid Model(s)
Alzheimer’s disease (familial)	Neural spheroid, cerebral organoid, cortical organoid [78,81,82]
Alzheimer’s disease (sporadic)	Cerebral organoid [83,84]
Angelman syndrome	Cortical organoid [85]
Autism spectrum disorder	Telencephalic organoid [86]
Bipolar disorder	Cerebral organoid [87]
Creutzfeldt–Jacob syndrome	Cerebral organoid [88]
Cytomegalovirus	Cerebral organoid [89]
Down’s syndrome (Trisomy 21)	Cerebral organoid [90]
Fragile X syndrome	Cortical organoid [91]
Frontotemporal dementia	Cerebral organoid [92]
Glioma	Cerebral organoid [93]
Hereditary spastic paraplegia	Cerebral organoid [94]
Huntington’s disease	Cortical organoid, cerebral organoid [95]
Japanese encephalitis	Cortical organoid [96]
Leber congenital amaurosis	Optic cup organoid [97]
Lissencephaly	Cortical organoid, cerebral organoid [98,99]
Parkinson’s disease	Midbrain organoid [100,101]
Pelizaeus–Merzbacher disease	Cortical spheroid [102]
Periventricular heterotopia	Cerebral organoid [103]
Progressive myoclonus epilepsy	Cerebral organoid [104]
Retinitis pigmentosa	Retinal organoid [105]
Rett syndrome	Cerebral organoid [106]
Sandhoff disease	Cerebral organoid [107]
Schizophrenia	Cerebral organoid [108,109]
Timothy syndrome	Forebrain organoid [110]
Tuberous sclerosis	Cortical spheroid [111]
ZIKV infection/microcephaly	Cerebral organoid [44,112]

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
