# Peer review of "Brain Organoids: Filling the Need for a Human Model of Neurological Disorder"

_biology, 2021, doi:10.3390/biology10080740_

Round 1

Reviewer 1 Report

This is a well-written review focusing on the development of the organoid technology and on its application as models for neurological diseases. Furthermore, the authors present advantages and disadvantages of the organoid model in comparison to other models of neurological diseases and provide a discussion about aspects of the organoid technology that need to be improved in future. While many aspects of this review were already discussed in detail in other reviews, the focus on neurological diseases gives it novelty. However, there is one point in this review that the authors have to address: The definition and the difference between “whole-brain organoids” and “single-region brain organoids” provided in this review is unclear. E.g., why are midbrain-specific organoids are counted as whole-brain organoids, while these organoids are specific to a single region (i.e., midbrain)? There are several reviews available that already defined different groups of organoids including “whole-brain organoids” and “region-specific organoids.” The authors could use these clearer definitions and cite the respective reviews.

Author Response

We agree with Reviewer 1 point, therefore we added a more clear definition of "whole" and "region" specific brain organoids in the "Complex 3D organoids and the use of patterning factors" paragraph.  Accordingly we moved in the region-specific brain organoid section the papers realted to mid-brain organoids.

Reviewer 2 Report

In this review manuscript, the authors discuss the emerging brain organoid technology, mainly placing special focus on two topics: one is the development of brain organoids companied with their strengths and weaknesses; the other one is the applications of different types of brain organoids as models of neurological disorders. The authors address the advancements of brain organoid technology   in mimicking the cellular diversity, cellular interconnectivity, and 3D structure of the CNS. Using the advanced brain organoid models, it’s possible to help us understand the underlying causes of neurodegenerative, neurodevelopmental, and neuropsychiatric disorders. I think the authors clearly introduced a promising technology that has produced great impact on the research of neurological disorders. The review article is logically organized, and the title is accurate and clearly identify the subject matter. The figures and tables are understandable and readable. I think this review article provides in-depth discussion on the significances and limitations of brain organoid, as well as a prospect to important research in the near future. I would recommend accepting this review article after considering some of my comments.

Major Comments:

  1. Could you please discuss a little bit of brain organoid’s applications on evaluating the acute and chronic toxin exposure?
  2. The cellular development and diversity are different between human brain and brain organoid. Could you please add some descriptions on this?
  3. The uniformity of brain organoid is important to decrease the result’s variation when studying the neurological disorders. Could you please overview some engineering methods on generating reliable brain organoids?

Minor Comment:

  1. The font format is supposed to be the same at line 163 ‘Self-renewal’, 164 ‘proliferation’, and 165 ‘multipotency’.

Author Response

We are thankful to Reviewer 2 for the positive comments.

The raised points are addressed below:

1) We have accordingly added new references in the discussion that refer to the application of toxins in brain organoids.

2) We mentioned the raised point in the discussion.

3) We expanded the paragraph "Advances in bioengineering" according to reviewer requests.